# Hull Shape Design Optimization with Parameter Space and Model Reductions, and Self-Learning Mesh Morphing

**Nicola Demo** * [iD] , **Marco Tezzele** [iD] , **Andrea Mola** [iD] and **Gianluigi Rozza** [iD]

Mathematics Area, mathLab, SISSA, Via Bonomea 265, I-34136 Trieste, Italy; marco.tezzele@sissa.it (M.T.); andrea.mola@sissa.it (A.M.); grozza@sissa.it (G.R.)
* Corresponding: nicola.demo@sissa.it

**Abstract:** In the field of parametric partial differential equations, shape optimization represents a challenging problem due to the required computational resources. In this contribution, a data-driven framework involving multiple reduction techniques is proposed to reduce such computational burden. Proper orthogonal decomposition (POD) and active subspace genetic algorithm (ASGA) are applied for a dimensional reduction of the original (high fidelity) model and for an efficient genetic optimization based on active subspace property. The parameterization of the shape is applied directly to the computational mesh, propagating the generic deformation map applied to the surface (of the object to optimize) to the mesh nodes using a radial basis function (RBF) interpolation. Thus, topology and quality of the original mesh are preserved, enabling application of POD-based reduced order modeling techniques, and avoiding the necessity of additional meshing steps. Model order reduction is performed coupling POD and Gaussian process regression (GPR) in a data-driven fashion. The framework is validated on a benchmark ship.

**Keywords:** shape optimization; reduced order modeling; high-dimensional optimization; parameter space reduction; computational fluid dynamics

## 1. Introduction

In the framework of parameterized partial differential equation (PDE) problems for engineering, reduced order models (ROMs) and optimization algorithms are two instruments that particularly benefit a synergic use. In several cases of engineering interest in which PDEs solution require considerable computational effort, ROMs enable in fact a remarkable reduction in the resources required for each calculation. There are of course several ways to reduce the dimensionality of discretized PDEs. The most naive approaches, such as coarsening the computational grids clearly have negative effects on the quality of the solutions. This is particularly true for problems characterized by complex physics and geometrical features, which in most cases require a very high number of degrees of freedom, ultimately resulting in expensive computations. In the context of an optimization algorithm execution, where many discretized PDE solutions must be computed, the overall computational load often becomes unaffordable. With only modest negative effects on the PDE solution accuracy, ROMs can be conveniently exploited to reduce the high dimensionality of the original discrete problem—to which we will herein refer to as full order model (FOM) or high fidelity model. ROM algorithms can be employed in several industrial design processes, and in particular to shape optimization, in which the objective of the computations is to find the best shape of a particular product or artifact. Such problems are in fact typically modeled through parametric PDEs, in which input parameters control the geometric features of the object at hand. ROMs efficiently approximate the numerical solution of the full order PDE with a suitable reduced surrogate, enabling drastic reduction in the computational burden of the overall optimization procedure.

There are of course several different algorithms which allow for an efficient reduction of the dimensionality of parametric problem. In the present contribution, we make

use of a data-driven approach based on proper orthogonal decomposition (POD) [1,2]. The equation-free nature of such method is often an essential feature in the industrial sector, where modularity and solvers encapsulation play a fundamental role. Indeed, the data-driven POD based ROM employed in the present optimization framework can be coupled with any PDE solver, as the data integration is enforced through the output of interest of the full order problem. Similar reduced methods have been proposed in [3,4] for the shape optimization of a benchmark hull, while additional improvements have been made coupling the ROM with active subspace analysis and different shape parameterization algorithms in [5–8]. We refer the readers interested in parametric hull shape variations using ROMs to [9], while we mention [10,11] for design-space dimensionality reduction in shape optimization with POD. Moving from hulls to propellers, data-driven POD has also been successfully incorporated in the study of marine propellers efficiency [12,13] as well as hydroacoustics performance [14].

A further aspect of novelty of the optimization framework proposed is related to the parameterization of the geometry. In typical shape optimization cycles, the surface of the object under study is deformed before the domain discretization takes place. Thus, the meshing phase is repeated for any deformed entity. Such approach has the clear advantage of allowing for good control of the quality of the computational grid produced for each geometry tested. Yet, it suffers of two main problems: (i) the meshing step may be expensive, both because its CPU time might be comparable to the resolution of the problem itself, and because mesh generation is specially intensive in terms of human operator hours required; (ii) a different mesh for each geometry does not allow for the application of POD or several other ROM approaches, which require that the mesh topology, as well as the number of degrees of freedom of the discretized problem, are conserved across all the shapes tested. Thus, assuming a generic deformation map is available, which morphs the initial object surface—not the grid—we exploit such deformation to train a radial basis function (RBF) interpolation that will extend the surface deformation to the nodes of the PDE volumetric mesh. In this sense, the method is capable to learn and propagate any deformation to a given mesh. Properly selecting the RBF kernel, we can then obtain a smooth deformation in all the discretized domain, not only ensuring that the overall parameterization map preserves the initial mesh quality but also its topology. We remark that in this work, free-form deformation (FFD) is used to deform the surface of the object under study. Yet, we stress that the RBF extension methodology is completely independent from the parameterization method chosen for the object geometry. A similar approach has been recently investigated in [15].

The optimization algorithm used in this work is the recently developed active subspaces extension of the classical genetic algorithm called active subspace genetic algorithm (ASGA) [16], which performs the mutation and cross-over steps on a reduced dimensional space for a faster convergence.

All the algorithms used in this work are implemented in open source software libraries [17–20], which we will briefly introduce in the discussions of the corresponding numerical methods. In Figure 1 we depicted an outline of the whole numerical pipeline we are going to present, emphasizing the methods and the softwares used. One of the main goals of this contribution it that of testing the full pipeline composed by data-driven POD ROM, combined FFD-RBF shape parameterization algorithm and ASGA optimizer on a problem that can be both meaningful to the ship hydrodynamics community and easily reproducible. For such reason, the test case considered is that of the DTC hull [21], for which online tutorials are available to run fairly accurate flow simulations in fixed sink and trim conditions. Since in such set up, the hull optimizing resistance is a trivial, zero volume hull, the DTC benchmark hull is here optimized based on the total resistance coefficient $C_t$. We organize the contribution as follows: Section 2 presents a deeper discussion about the parameterization of the object and of the computational grid; Section 3 describes the full order model and the reduced order one, while Section 4 is devoted to an algorithmic discussion about the optimization algorithm and its supporting mathematical tools. The

final Sections 5 and 6, show the numerical results obtained and present the conclusive summary, respectively.

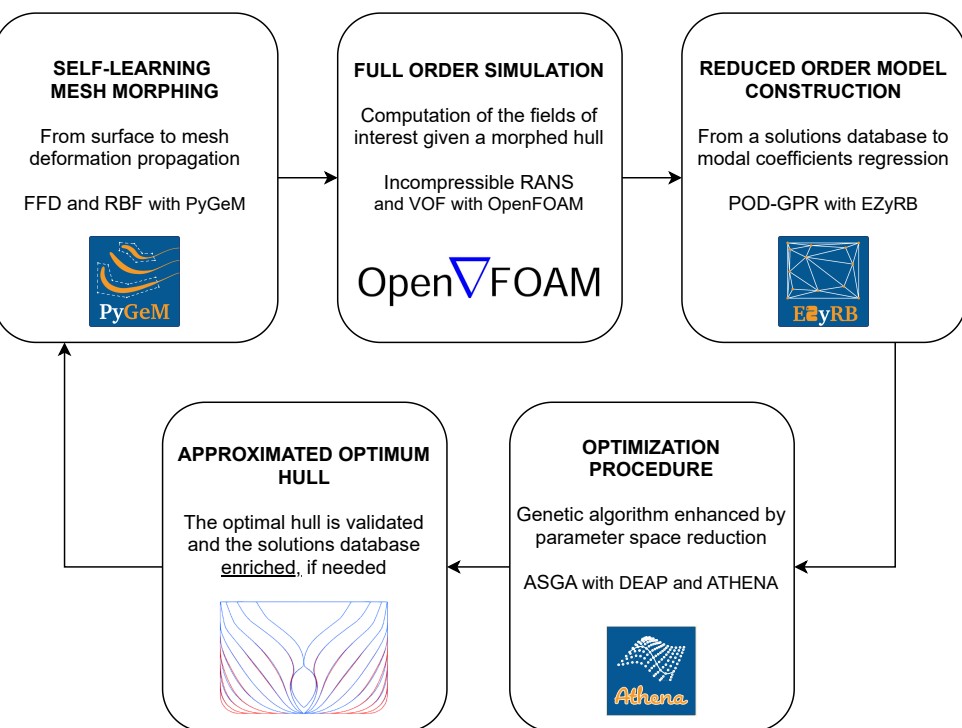

**Figure 1.** Illustration of the key steps of the proposed optimization pipeline with the methods and the softwares used.

## 2. Shape and Grid Parameterization

Whenever industrial design processes as the ones discussed in this work are aimed at improving, among other aspects, the geometric features of a particular artifact, a shape parameterization algorithm is a cornerstone of the whole optimization pipeline. Optimization tools, as well as the non-intrusive model reduction techniques employed in the present investigation, are in fact based on the parameterized PDEs paradigm introduced in the previous section. In such framework, a set of geometric input parameters affects the output of a parametric PDE through the deformation of its domain geometry. Thus, the shape parameterization algorithm role is that of mapping the variation of a set of numerical parameters, to the corresponding deformation of the PDE domain geometry. In other words, since optimization tools are mathematical algorithms which must be fed with numbers, the shape parameterization algorithms translate shape deformations into variations of the numeric quantities they need.

*How to Combine Different Shape Parametrization Strategies*

In this work, we make combined use of two general purpose shape parameterization algorithms to deform the three dimensional geometry of a ship hull, and accordingly update the volumetric grid used for ship hydrodynamics simulations in a fully automated fashion. More specifically, free form deformation (FFD) is first used to generate a family of deformations of the surface of a base hull. In a second step, radial basis functions (RBF) interpolation is used to propagate the hull surface deformation to the internal nodes of the fluid dynamic simulation computational grid. For visual reference, Figure 2 depicts the side view (on the left) and front view (on the right) of a container ship hull bow region. In the picture, several sections perpendicular to the hull longitudinal axis are indicated by red lines.

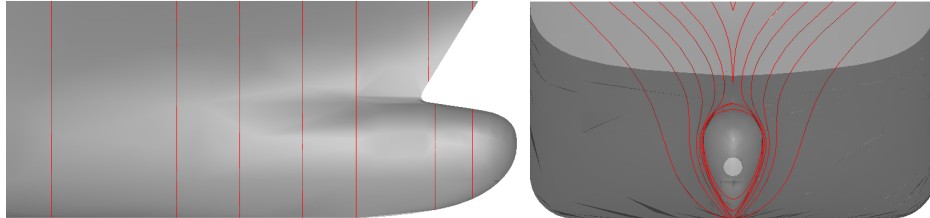

**Figure 2.** Side view (**left**) and front view (**right**) of a typical container ship hull bow region.

Despite an extensive discussion of FFD and RBF theoretical foundations is clearly beyond the scope of the present contribution, this section will introduce the key concept upon which both algorithms are based and describe their combined deployment in the framework of our optimization pipeline.

The first shape parameterization algorithm applied in this work is the free form deformation [22–24]. As mentioned, it is a general purpose algorithm, designed to be applied to arbitrarily shaped geometries. FFD is fundamentally made up of three different geometrical transformations, as illustrated in Figure 3. The first transformation $\psi$ maps the physical domain $\Omega$ into a reference domain $\widehat{\Omega}$. In such domain, a lattice of points is generated, and are used as the control points of a set of smooth shape functions such as the Bernstein polynomials used in this work. Thus, once a displacement is prescribed to one or more of the control points in the lattice, the shape functions are used to propagate such displacement to all the points in the reference domain $\Omega$. The smooth displacement field obtained, is the second and most important transformation $\widehat{T}$ in the FFD process. In the third, final step, the deformed reference domain is mapped back into the physical one by means of $\psi^{-1}$ to obtain the resulting morphed geometry.

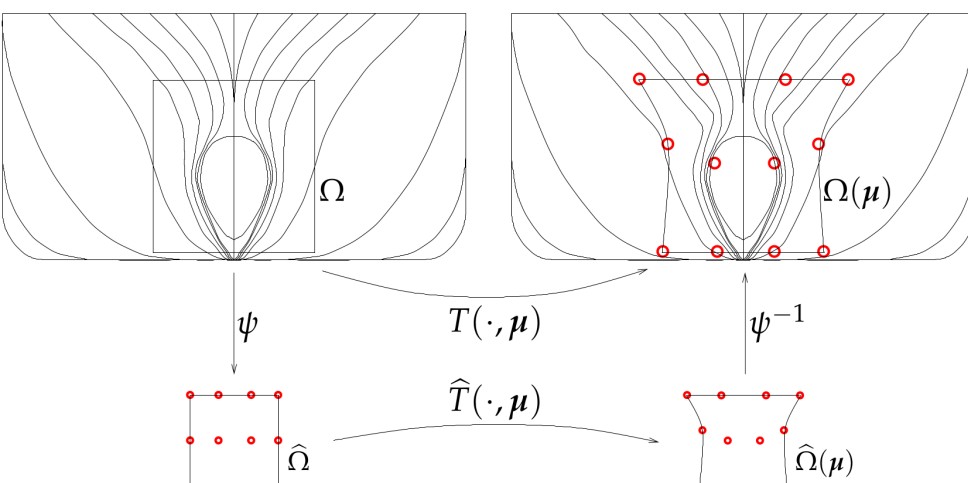

**Figure 3.** A two dimensional sketch of the free form deformation (FFD) procedure applied to the surface of a container ship hull, including the three transformations $\psi$, $\widehat{T}(\cdot, \boldsymbol{\mu})$ and $\psi^{-1}$ composing the process.

The current description suggests that the parameters $\boldsymbol{\mu}$ of the final FFD map $T(\cdot, \boldsymbol{\mu})$ are the displacements prescribed to one or more of the lattice control points. The procedure can account for both a variable number of lattice points and of displaced control points. For such reason, FFD deformations can be built with an arbitrary number of parameters.

We point out that the FFD algorithm results in a displacement law for each 3D space point within the control points lattice. Thus, it can be readily deployed to deform shapes specified through surface triangulations (such as STL geometries) and surface grids in general. In addition, it can be also used to directly deform volumetric grids used for fluid dynamic simulations. Yet, mainly for practical reasons, in this work we only make use of FFD to deform the STL surface triangulation describing the hull geometry. In fact, we

must point out that if FFD has to be used to modify the volumetric mesh used for CFD simulations, the control points lattice dimensions must be much bigger than those needed when only deforming the hull surface, leading to infeasible optimization procedures. This is due to the fact that when deforming volumetric meshes, it is often convenient to distribute the deformations over a high number of cells, rather than concentrating all the displacements in a very confined region in which cells can get distorted or even inverted. But because FFD only affects points located within the control points lattice, this means that the latter must extend for a bigger volume. In addition, to maximize the volumetric mesh quality, the user must include more control points in the lattice to make sure that different deformation magnitudes are imposed in regions close to the hull and far from it. Such manual control over the local mesh deformation can often become quite cumbersome.

For such reasons, after the hull surface mesh has been modified by means of FFD, we resort to RBF to propagate the hull boundary displacements to the internal nodes of the volumetric mesh for CFD simulations. In a broader sense, RBF is an interpolation algorithm, in which linear combinations of radial bases are used to approximate a function with values prescribed only in a finite number of points, in every point of a domain. In the case of interest, the displacement field function prescribed on the points of the hull surface must be interpolated in the positions corresponding to every node of the volumetric mesh. Thus, the displacement obtained from the $m$ surface nodes original position $\{s_1, \ldots, s_m\}$ and the corresponding displaced position $\{s'_1, \ldots, s'_m\}$ must be interpolated at the positions $\{v_1, \ldots, v_n\}$ of the $n$ volumetric mesh nodes. Such interpolation reads

$$d(x) = \sum_{j=1}^{m} w_j \varphi_j(x), \tag{1}$$

where the radial bases $\varphi_j(x) = \varphi_j(||x - x_j||)$ are functions that only depend on the distance between evaluation point $x$ and control point $x_j$. The weights $w_j$ are computed by imposing the interpolation constraints $d(s_i) = s'_i - s_i$, after a radial basis has been centered at every constrained point $(x_j = s_j)$. This results in the linear system

$$AX = B, \tag{2}$$

where

$$A = \begin{bmatrix} \varphi_1(s_1) & \cdots & \varphi_1(s_m) \\ \vdots & \ddots & \vdots \\ \varphi_m(s_1) & \cdots & \varphi_m(s_m) \end{bmatrix}, \quad X = \begin{Bmatrix} w_1 \\ \vdots \\ w_m \end{Bmatrix}, \quad B = \begin{Bmatrix} s'_1 - s_1 \\ \vdots \\ s'_m - s_m \end{Bmatrix}. \tag{3}$$

Linear system (2) is solved in a pre-processing phase, and the weights computed are then used to compute the displacement of every node of the volumetric mesh by means of Equation (1). The latter operation can be conveniently carried out in a parallel fashion, and is highly efficient. On the other hand, $A$ is a full $m \times m$ matrix which can make the solution of system (2) quite time and memory demanding when a large number of RBF control points are considered. That is why, in some cases only a portion of the surface mesh nodes are used as RBF control points, which limits the computational cost more than linearly, and in most cases has only modest effect on the morphing accuracy.

Both the FFD and RBF algorithms briefly described in this section have been implemented in the Python library for geometrical morphing PyGeM [17], which has been used to produce all the deformed geometries and computational grids used in this work. An example of the RBF application to volumetric mesh morphing described in this paragraph is presented in Figure 4. The figure illustrates all the steps involved in the procedure, which starts with (a) a first volumetric mesh around the hull, and (b) a surface mesh on the hull surface. In step (c) the latter mesh is then deformed and (d) the surface mesh displacement field is finally used to feed the RBF algorithm and propagate the boundary motion to the internal volumetric mesh nodes. As it can be appreciated in the illustration, to avoid

distortion of the volumetric mesh symmetry plane, the surface mesh must include both sides of the hull. In the present work, the deformation of the surface mesh has been carried out by means of FFD. Yet, we remark that any deformation law which results in a one to one correspondence between original and deformed surface grids can be propagated to the nodes of the volumetric mesh with RBF interpolation.

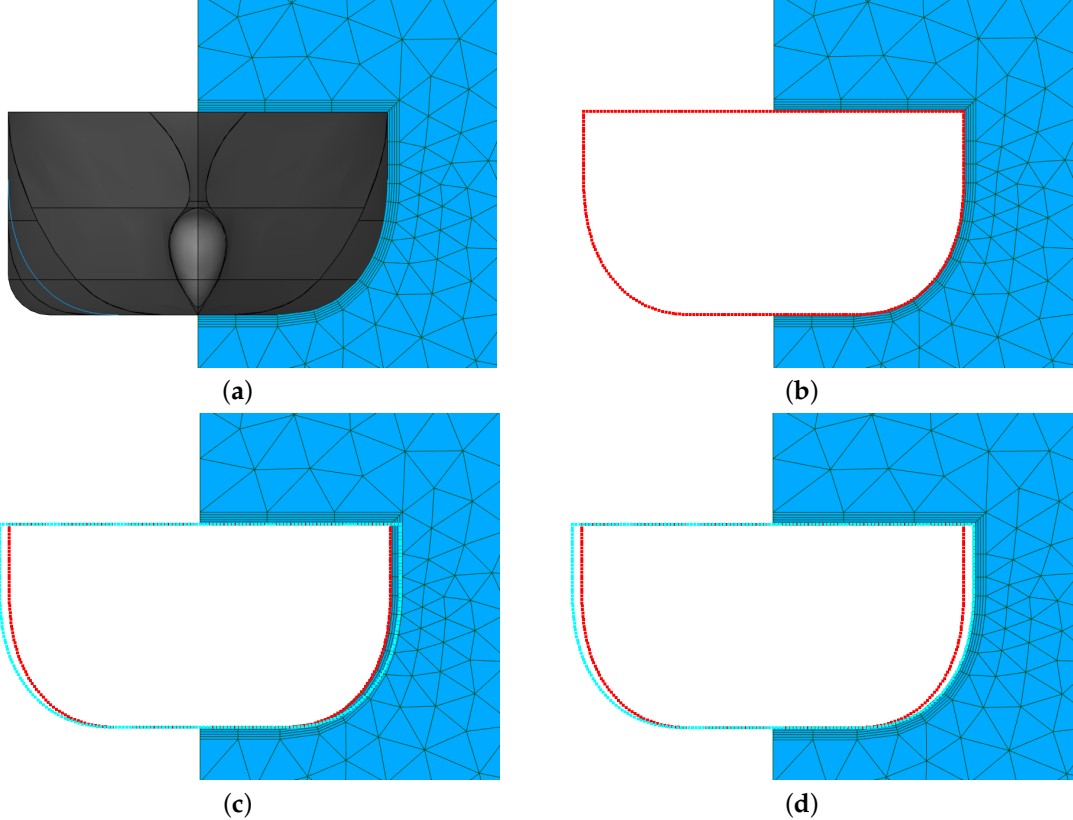

**Figure 4.** A section view example illustrating the radial basis functions (RBF) morphing steps carried out to propagate the hull surface deformations to a volumetric mesh for ship hydrodynamics simulations.

## 3. The Mathematical Model for Incompressible Fluids

The computational gain of the proposed pipeline is obtained by using a model order reduction based on proper orthogonal decomposition (POD) to approximate the solution of the parametric PDEs describing the studied phenomenon. This technique assumes an initial solutions database produced by solving the full order model (FOM), for some values of the parameters. We refer to such solutions as high-fidelity solutions, or *snapshots*. Depending on the intrusiveness of the reduced order method, also the discrete operators of the numerical problem can be required. In this contribution, we propose a non-intrusive approach, constructing a reduced order model (ROM) within a data driven setting using the FOM snapshots and the corresponding parameter values (described in Section 2). This allows a modular structure where any numerical solver, also commercial, can be adopted, since the ROM relies only on input and output couples.

The following paragraphs present the full order model used in this work and the ROM constructed with it. We briefly describe the incompressible Reynolds Averaged Navier-Stokes (RANS) equations and its numerical solution in a finite volume framework, then we proceed with an algorithmic analysis of the proper orthogonal decomposition with Gaussian process regression (POD-GPR).

### 3.1. The Full Order Model: Incompressible Rans

The FOM used in this work is the Reynolds Averaged Navies–Stokes (RANS) model complemented by a Volume of Fluid (VOF) front capturing method to deal with the multi phase nature of the fluid surrounding the hull. The resulting govern equations are discretized by means of a Finite Volumes (FV) strategy implemented in the open source library openFOAM [25]. Such mathematical and numerical setup is nowadays adopted in many industrial naval contexts thanks to its robustness and accuracy. The test case considered is one of the tutorials of the library, which is designed to reproduce the DTC experiments reported in reference [21]. We here provide a minimal overall description of the model. We refer to the original documentation of the library for all the numerical and technical details.

The RANS equations model the turbulent incompressible flow, while the volume of fluid (VOF) technique [26] is applied to handle the biphase nature of the fluid (water and air). The equations governing our system are the following

$$\begin{cases} \frac{\partial \bar{u}}{\partial t} + (\bar{u} \cdot \nabla)\bar{u} - \nabla \cdot (\tilde{u} \otimes \tilde{u}) = -\frac{1}{\rho}\nabla \bar{p} + \nabla \cdot \nu \nabla \bar{u} + g, \\ \nabla \cdot \bar{u} = 0, \\ \frac{\partial \alpha}{\partial t} + \nabla \cdot (\bar{u}\alpha) = 0, \end{cases} \tag{4}$$

where $\bar{u}$ and $\tilde{u}$ refer to the mean and fluctuating velocity after the RANS decomposition, respectively, $\bar{p}$ denotes the mean pressure, $\rho$ is the density, $\nu$ the kinematic viscosity, and $\alpha$ is the discontinuous variable belonging to interval $[0, 1]$ representing the fraction of the second flow in the infinitesimal volume. Finally, vector $g$ represents the body accelerations associated with gravity.

The first two equations are the continuity and momentum conservation, where the new term, the Reynolds stresses tensor $\tilde{u} \otimes \tilde{u}$, have to be modeled with additional equations in order to close the system. Among all the turbulence models available in literature, we use the SST$k - \omega$ turbulence model [27]. The third equation represents the transport of the VOF variable $\alpha$. Such variable controls also the density $\rho$ and the kinematic viscosity $\nu$, since they are defined using an algebraic formula expressing them as a convex combination of the corresponding properties of the two flows such that

$$\rho = \alpha \rho_{\text{air}} + (1 - \alpha)\rho_{\text{water}}, \qquad \nu = \alpha \nu_{\text{air}} + (1 - \alpha)\nu_{\text{water}}. \tag{5}$$

To compute the steady solution in a discrete environment, we apply the finite volume (FV) approach. We set a pseudo–transient simulation, applying a first order implicit local scheme for the temporal discretization, while for the spatial scheme we apply the linear upwind one. Regarding the software, as mentioned the simulation is carried out using the C++ library OpenFOAM [25].

### 3.2. The Reduced Order Model: POD-GPR

POD is a linear dimensional reduction technique capable to construct a reduced order model from a set of high-fidelity snapshots. Such space is spanned by (typically few) basis functions, that are computed by minimizing the error between the original snapshots and their orthogonal projection [28]. In a parametric context, it enables—provided a proper set of parameter samples—the possibility to approximate the solution manifold in a very efficient way. Formally, we define the set of parameters $\{\mu_i\}_{i=1}^{M}$ such that $\mu_i \in \mathbf{P} \subset \mathbb{R}^p$ for $i = 1, \ldots, M$. For each parameter, the solution is computed using the FOM. Let $\mathcal{N}$ be number of degrees of freedom of the full simulation, we obtain the solutions $x_i \in \mathbb{X}_i^{\mathcal{N}}$ for $i = 1, \ldots, M$. Since the finite volume space is created only once and then it is deformed, all

the geometric configurations have the same dimensionality even if they belong to different spaces. The vectorial solutions are arranged as columns of the snapshots matrix, such that

$$\mathbf{X} = \begin{bmatrix} | & \cdots & | \\ \mathbf{x}_1 & \cdots & \mathbf{x}_M \\ | & \cdots & | \end{bmatrix} \in \mathbb{R}^{\mathcal{N} \times M}. \tag{6}$$

The basis of the POD space, composed by the so called POD modes, is computed using the singular value decomposition (SVD) of the snapshots matrix $\mathbf{X} = \mathbf{U \Sigma V}^*$. The unitary matrix $\mathbf{U} \in \mathbb{R}^{\mathcal{N} \times M}$ contains the left-singular vectors of $\mathbf{X}$, which are the POD modes. Moreover the diagonal matrix $\mathbf{\Sigma} = \mathrm{diag}(\lambda_1, \ldots, \lambda_M)$, where $\lambda_1 \geq \lambda_2 \geq \ldots \geq \lambda_M$, contains the singular values, which indicate the energetic contribution of the corresponding modes. By looking at the spectral decay we can retain the first $N$ most energetic modes, which span the optimal space of dimension $N$.

Such basis can be exploited in a Galerkin projection framework [29–31], in an hybrid framework combining data-driven methods with projection [32,33], or used to project onto the reduced space the initial snapshots. Thus we can approximate the snapshots $\mathbf{x}_j$ as a linear combination of the modes as

$$\mathbf{x}_j = \sum_{i=1}^{M} \mathbf{c}_j^i \boldsymbol{\psi}_i \approx \sum_{i=1}^{N} \mathbf{c}_j^i \boldsymbol{\psi}_i \quad \text{for } j = 1, \ldots, M, \tag{7}$$

where $\boldsymbol{\psi}_i$ refers to the $i$-th POD mode. The coefficients $\mathbf{c}_j^i$ of the linear combination represent the low-dimensional solution and are usually called *modal coefficients*. Using the matrix notation, to compute such coefficients it is sufficient a matrix multiplication $\mathbf{C} = \mathbf{U}_N^T \mathbf{X}$, where the columns of $\mathbf{C}$ are the vectors $\mathbf{c}^j \in \mathbb{R}^N$ for $j = 1, \ldots, N$, the matrix $\mathbf{U}_N \in \mathbb{R}^{\mathcal{N} \times N}$ contains the first $N$ POD basis and the superscript $T$ indicates the matrix transpose.

The new pairs $(\boldsymbol{\mu}_i, \mathbf{c}_i)$, for $i = 1, \ldots, M$, we can be exploited in order to find a function $f : \mathbf{P} \to \mathbb{R}^N$ capable to predict the modal coefficients for untested parameters. Several options are available in literature to reach this goal: for instance $n$-dimensional linear interpolator [34,35], radial basis functions (RBF) interpolator [36], artificial neural networks [37], Gaussian process regression [38,39]. As anticipated, in this work we apply a GPR [40], fitting the distribution of the modal coefficients with a multivariate Gaussian distribution, such that

$$f(\boldsymbol{\mu}) \sim \mathrm{GP}(m(\boldsymbol{\mu}), K(\boldsymbol{\mu}, \boldsymbol{\mu})), \tag{8}$$

where $m(\cdot)$ and $K(\cdot, \cdot)$ indicate the mean and the covariance of the distribution, respectively. Given a covariance function, an optimization step is required to set the corresponding hyperparameters. In this contribution we use the squared exponential covariance defined as $K(x_i, x_j) = \sigma^2 \exp\left(-\frac{\|x_i - x_j\|^2}{2l}\right)$. Once the hyperparameters ($\sigma$ and $l$) of the covariance kernel have been fit to the input dataset, we can query such distribution to predict the new modal coefficients. Finally the modal coefficients are projected back to the high-dimensional vector space $\mathbb{R}^{\mathcal{N}}$ using (7). It is easy to note the differences from the computational point of view between FOM and ROM—whereas in the full order model it is required to solve a non-linear problem of dimension $\mathcal{N}$, in the reduced order model to predict the solution we just need to query a distribution and perform a matrix multiplication. From the computational perspective, in fact the cost of the ROM is mainly due to its construction and not to the prediction phase: relying on the SVD, the method shows an algorithmic complexity of $\mathcal{O}(\min(\mathcal{N}, M) \mathcal{N} M)$. Thus, dealing with complex FOM as the one presented in this work, POD space construction can be neglected in the overall computational need.

On the technical side, we construct and exploit the POD-GPR model using EZyRB [19], an open source Python package which deals with several data-driven model order reduction techniques, exploiting the library GPy [20] for the GPR implementation.

## 4. Optimization Procedure with Built-in Parameters Reduction

In this work we make use of the active subspaces extension of the genetic algorithm (ASGA) introduced in [16]. Such optimization method has been selected as it outperforms standard GA, especially when high-dimensional target functions are considered. Its performance have been proved both for classical academic benchmark functions and for industrial CFD test cases.

The following sections report a description of both the classical genetic algorithm and the active subspaces technique main features. Finally, we will discuss how the two algorithms have been combined to obtain an efficient optimization procedure.

### 4.1. Genetic Algorithm

Genetic algorithm (GA) is an optimization algorithm, first introduced by Holland in [41]. Inspired by natural selection, it falls into the category of population based search algorithms. For a detailed discussion of the method and its several modifications we refer the interested reader to [42–44]. Here, we briefly present the simplest genetic algorithm, which is composed by three fundamental steps: *selection*, *reproduction*, and *mutation*. Such phases are illustrated in Figure 5—which also includes yellow boxes which will be discussed in the following sections.

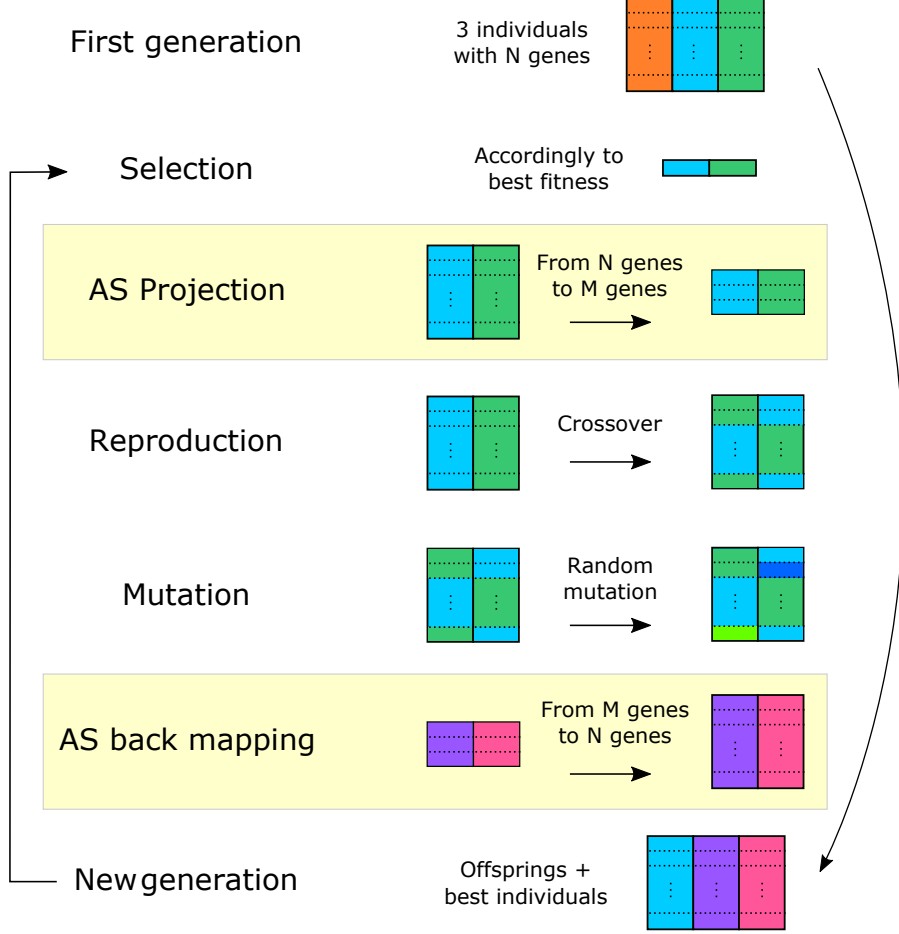

**Figure 5.** Active subspaces-based genetic algorithm scheme. The main step of the classical genetic algorithm (GA) are depicted from top to bottom. The yellow boxes represent projections onto and from lower dimension active subspace. Thus, they are specific to ASGA.

The algorithm starts with a random population $\mathcal{S}_0$ composed of $T$ individuals, each one having $r$ genes. In the selection step the individuals with the best fitness value, for instance $\mathcal{S}_0^{(1)}$ and $\mathcal{S}_0^{(2)}$, are retained. During the reproduction phase, an offspring $Q$ is

produced from these two individuals with a crossover probability $P_C$. Then, in the last step $Q$ undergoes a mutation with probability $P_M$, generating $Q'$. This new offspring $Q'$ is added in the new population $\mathcal{S}_1$ together with the best individuals of $\mathcal{S}_0$. The three steps are repeated until a predetermined computation budget is reached.

### 4.2. Active Subspaces

The active subspaces (AS) [45–47] property is an emerging technique for dimension reduction of parameterized problems. Let us initially assume that the input/output relationship of the problem under study is represented by function $f(\boldsymbol{\mu}) : \Omega \subset \mathbb{R}^n \to \mathbb{R}$. The reduction is performed by computing a linear transformation of the original parameters $\boldsymbol{\mu}_M = \mathbf{A}\boldsymbol{\mu}$, in which $\mathbf{A}$ is an $M \times n$ matrix, and $M < n$. In the last years AS has been extended to vector-valued output functions [46], and to nonlinear transformations of the input parameters using the kernel-based active subspaces (KAS) method [48]. AS has been also coupled with reduced order methods such as POD-Galerkin [49] in cardiovascular studies, and POD with interpolation [50] and dynamic mode decomposition [51] for CFD applications. Application to multi-fidelity approximations of scalar functions are also presented in [52,53].

The matrix $\mathbf{A}$ is computed based on the second moment matrix $\mathbf{C}$ of the target function $f$ gradient. The latter matrix is defined as

$$\mathbf{C} := \mathbb{E}\left[\nabla_{\boldsymbol{\mu}}f\,\nabla_{\boldsymbol{\mu}}f^T\right] = \int (\nabla_{\boldsymbol{\mu}}f)(\nabla_{\boldsymbol{\mu}}f)^T\rho\,d\boldsymbol{\mu}, \tag{9}$$

where with $\mathbb{E}[\cdot]$ we denote the expected value, $\nabla_{\boldsymbol{\mu}}f \equiv \nabla f(\boldsymbol{\mu}) \in \mathbb{R}^n$, and $\rho : \mathbb{R}^n \to \mathbb{R}^+$ is a probability density function representing the uncertainty in the input parameters. The gradients appearing in $C$ are typically approximated [45] with local linear models, global linear models, GP regression, or finite difference. The second moment matrix $\mathbf{C}$ is constructed with a Monte Carlo procedure. We proceed by decomposing the uncentered covariance matrix as $\mathbf{C} = \mathbf{W}\Lambda\mathbf{W}^T$, where $\Lambda$ is the diagonal eigenvalues matrix (arranged in descending order) and $\mathbf{W}$ is the orthogonal matrix containing the corresponding eigenvectors. To bound the error on the numerical approximation associated with Monte Carlo simulations, we make use of the gap between the eigenvalues. Looking at the energy decay, we can select a scalar $M < n$ and decompose $\Lambda$ and $\mathbf{W}$ as

$$\Lambda = \begin{bmatrix} \Lambda_1 & \\ & \Lambda_2 \end{bmatrix}, \quad \mathbf{W} = [\mathbf{W}_1 \quad \mathbf{W}_2], \quad \mathbf{W}_1 \in \mathbb{R}^{n \times M}, \tag{10}$$

where $M$ is the dimension of the active subspace–which can also be prescribed a priori. The decomposition described is exploited to map the input parameters onto a reduced space. Thus, the principal eigenspace corresponding to the first $M$ eigenvalue defines the *active subspace* of dimension $M$. In particular we define the active variable as $\boldsymbol{\mu}_M := \mathbf{W}_1^T\boldsymbol{\mu} \in \mathbb{R}^M$ and the inactive variable as $\boldsymbol{\eta} := \mathbf{W}_2^T\boldsymbol{\mu} \in \mathbb{R}^{n-M}$.

Exploiting the higher efficiency of most interpolation strategy in lower dimensional spaces, we can now approximate $f$ using a response surface over the active subspace, namely

$$g(\boldsymbol{\mu}_M = \mathbf{W}_1^T\boldsymbol{\mu}) \approx f(\boldsymbol{\mu}), \qquad \boldsymbol{\mu}_M \in \mathcal{P} := \{\mathbf{W}_1^T\boldsymbol{\mu} \mid \boldsymbol{\mu} \in \Omega\}, \tag{11}$$

where $\mathcal{P}$ is the polytope in $\mathbb{R}^M$ (the ranges of the parameters are intervals) defined by the AS.

The active subspaces technique and several other methods for parameter spaces reduction are implemented in the ATHENA[1] Python package [18].

---

[1] Freely available at https://github.com/mathLab/ATHENA (accessed date 1 January 2020).

### 4.3. Active Subspaces-Based Genetic Algorithm

We enhance the classical GA by adding two fundamental steps before the reproduction and after the mutation phase. These involve the application of the projection of the current population onto its active subspace, given a prescribed dimension. So, the idea is to perform the crossover and the random mutation in the smaller dimension space. Such space in fact only includes the directions in which the highest variation of the fitness function $f$ is observed.

By a mathematical standpoint, we add the following operations to the GA: let $\mathbf{W}_1$ be the eigenvectors defining the active subspace of the current population, say $\mathcal{S}_0$. We project its best individuals onto the current active subspace with

$$s_0^{(1)} = \mathbf{W}_1^T \mathcal{S}_0^{(1)}, \qquad s_0^{(2)} = \mathbf{W}_1^T \mathcal{S}_0^{(2)}, \tag{12}$$

where $s_0^{(1)}$ and $s_0^{(2)}$ are the reduced individuals. The reproduction and mutation steps are performed as usual. The only difference is that in the described framework they conveniently are carried out within a smaller dimension space, where reduced number of genes is exploited for speed up purposes. After these phases are completed, we obtain the offspring $q$ and $q'$, respectively. Finally, the back mapping from the active subspace to the full space is performed by sampling the inactive variable $\boldsymbol{\eta}$ in order to obtain

$$Q' = \mathbf{W}_1 q' + \mathbf{W}_2 \boldsymbol{\eta}, \qquad \text{with } -\mathbf{1} \leq Q' \leq \mathbf{1}, \tag{13}$$

where $\mathbf{1}$ denotes a vector with all components equal to 1—the original parameters are usually rescaled in $[-1, 1]^n$ before applying AS—. We remark that there is in principle the possibility that multiple points in the full space are mapped onto the same reduced point in the active subspace. Hence, the number $B$ of individuals resulting from the back mapping is an hyperparameter which can be prescribed a priori. For the specifics about this procedure please refer to [16]. In Figure 5 we emphasized with yellow boxes the new fundamental steps represented by Equations (12) and (13). For the actual implementation of the genetic algorithm part we used DEAP [54].

### 5. Numerical Results

In this section, we describe the application of the proposed optimization pipeline to the DTC hull surface. Table 1 shows the main particulars in the design loading condition at model scale (which is set to 1:59.407). This will provide a test case which closely simulates a typical workflow for industrial hull design problems. Figure 6 shows the original CAD geometry of the hull used in this work, where we marked 21 longitudinal sections which divide the ship into 20 equispaced chunks. Such 21 slices will be referred to as *sections* during the results discussion, and are numbered from 1 to 21 going from the ship stern to its bow.

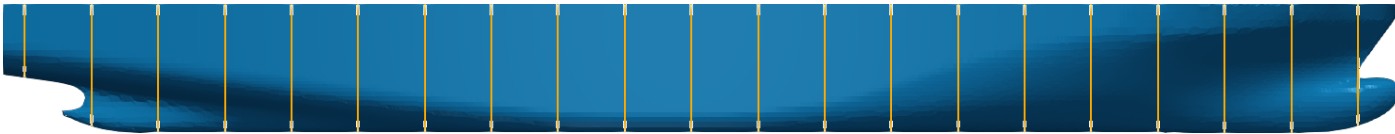

**Figure 6.** The surface of the DTC hull. The highlighted sections divide the ship into 20 equispaced chunks at the free-surface level.

**Table 1.** Main quantities of the DTC at scale model.

| Quantity | Value |
| --- | --- |
| Length between perpendiculars $L_{pp}$ [m] | 5.976 |
| Waterline breadth $B_{wl}$ [m] | 0.859 |
| Draught midships $T_m$ [m] | 0.244 |
| Volume displacement $V$ [m$^3$] | 0.827 |
| Block coefficient $C_B$ | 0.661 |

The structure of this section mirrors that of the whole article, reporting the intermediate results of all the methods employed throughout the optimization pipeline.

### 5.1. Self-Learning Mesh Morphing Parameters

To set up the FFD hull surface deformation, we position the control points lattice in order to control the immersed part of the ship prow region. The equispaced control points are positioned as follows:

- *x* **axis:** 7 points layers located on sections 10, 12, 14, 16, 18, 20, 22;
- *y* **axis:** 11 points layers that cover the whole hull beam, with the second and the second-to-last positioned on the lateral walls of the ship;
- *z* **axis:** 7 points layers that cover the whole hull draft, aligning the 2nd and the 5th of them to the hull bottom and to the waterline, respectively.

As can be appreciated by the values reported, to distribute the FFD control points, we have made use of an additional 22nd virtual section located ahead of the bow. The motion of the $7 \times 11 \times 7 = 539$ points is governed by only 10 parameters, which are described in Table 2. We point out that the displacement of all the boundary points in the $x$ and $z$ direction is set to zero so as to enforce surface continuity. In addition, the displacement of the points on the internal $x$ and $z$ layers closest to the boundary ones is also set to zero so as to enforce continuity of all surface derivatives. Finally, the hull symmetry along $y$ direction is ensured by selecting symmetric values for parameters associated to $x$ and $z$ displacements, as well as antisymmetric values for parameters associated to $y$ displacements (the latter points are also indicated in the table by the corresponding footnote).

**Table 2.** FFD control points displacement. The indices refer to the relative position of the points within the lattice. The layers order, which starts from 0, is maintained consistent with the reference system. The intervals indicated by the—symbol are inclusive.

| Lattice Points | | | Parameter | Displacement Direction |
|---|---|---|---|---|
| Index $x$ | Index $y$ | Index $z$ | | |
| 2 | 0 | 2–4 | $\mu_0$ | $x$ |
| 2 | 10 | 2–4 | $\mu_0$ | $x$ |
| 3 | 0 | 2–4 | $\mu_1$ | $x$ |
| 3 | 10 | 2–4 | $\mu_1$ | $x$ |
| 4 | 0 | 2–4 | $\mu_2$ | $x$ |
| 4 | 10 | 2–4 | $\mu_2$ | $x$ |
| 4 | 2–4 | 2 | $\mu_3$ | $y$ |
| 4 | 6–8 | 2 | $-\mu_3{}^2$ | $y$ |
| 4 | 2–4 | 3 | $\mu_4$ | $y$ |
| 4 | 6–8 | 3 | $-\mu_4{}^2$ | $y$ |
| 4 | 2–4 | 4 | $\mu_5$ | $y$ |
| 4 | 6–8 | 4 | $-\mu_5{}^2$ | $y$ |
| 3 | 2–4 | 2 | $\mu_6$ | $y$ |
| 3 | 6–8 | 2 | $-\mu_6{}^2$ | $y$ |
| 5 | 2–4 | 3 | $\mu_7$ | $y$ |
| 5 | 6–8 | 3 | $-\mu_7{}^2$ | $y$ |
| 4 | 0–1 | 2 | $\mu_8$ | $z$ |
| 4 | 9–10 | 2 | $\mu_8$ | $z$ |
| 5 | 0 | 3 | $\mu_9$ | $z$ |
| 5 | 10 | 3 | $\mu_9$ | $z$ |

Once defined the geometric parameters $\boldsymbol{\mu} = [\mu_0, \ldots, \mu_9]$, we set the parametric space to $\mathbf{P} = [-0.2, 0.2]^{10}$. The parameter space boundary values are selected so as to obtain feasible deformations from an engineering point of view and, at same time, to explore a large

variety of possible shapes. Figure 7 shows the two "extreme" hull deformations, obtained setting all the parameters equal to the lower and upper bound of the space, respectively.

The FFD deformation of the hull points has been extended to the nodes of the volumetric grid for the CFD simulations making use of the Beckert-Wendland radial basis function kernel [55], defined as follows

$$\varphi_j(||\boldsymbol{x} - \boldsymbol{x}_j||) = \left(1 - \frac{||\boldsymbol{x} - \boldsymbol{x}_j||}{R}\right)_+^4 \left(1 + 4\frac{||\boldsymbol{x} - \boldsymbol{x}_j||}{R}\right), \tag{14}$$

where $R > 0$ is a prescribed finite radius and the $(\cdot)_+$ symbol indicates the positive part.

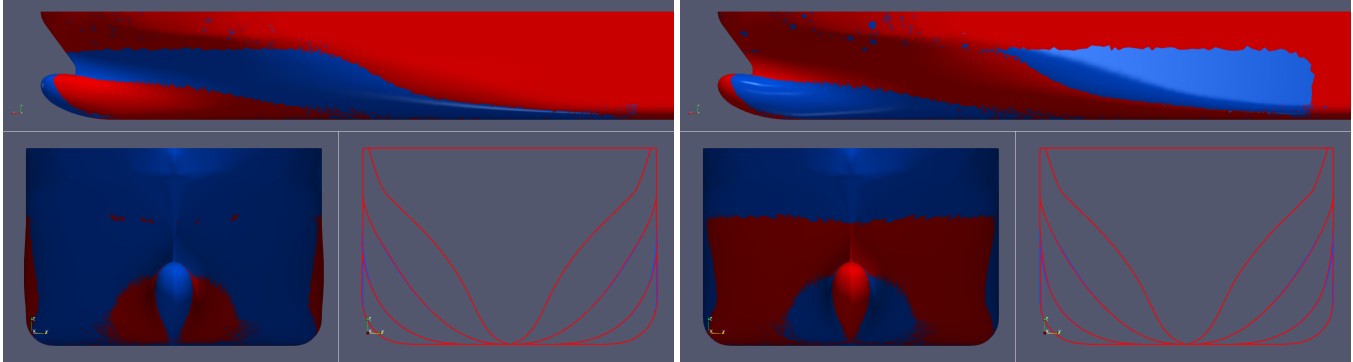

**Figure 7.** Visual examples of hull deformation with $\boldsymbol{\mu} = [-0.2]^{10}$ (on **left**) and $\boldsymbol{\mu} = [0.2]^{10}$ (on **right**). The red surface refers to the deformed ships, while the blue one is the original hull.

The output of the OpenFOAM library checkMesh utility has been used to assess the quality of the grids obtained with the combined FFD/RBF methodology. Figure 8 presents some of the main quality indicators of the 200 meshes generated for the present campaign, as computed by checkMesh. In particular, the indicators considered are minimum face area (top left plot), minimum cell volume (top right plot), maximum mesh non-orthogonality (bottom left plot) and average mesh non-orthogonality (bottom right plot). In all the diagrams, the vertical axis refers to the mesh quality indicator considered, while the variable associated with the horizontal axis is the index corresponding to each of the 200 volumetric meshes produced for the simulation campaign.

The minimum face area and minimum cell volume results indicate that the morphing procedure does not produce negative cells or faces which would impair the simulations. In fact, the average of both indicators across the 200 grids produced is extremely close to the corresponding value of the original grid. The lowest value of minimum face area observed in the 200 grids generated is less than 0.1% off the original value, while the lowest value of minimum cell volume observed is merely 0.01% off the original mesh minimum cell volume. Such trend is confirmed by the maximum non-orthogonality values reported in the bottom left diagram. In the plot, is possible to appreciate that the average over the 200 grids produced falls exactly on value of the original mesh, and the highest difference with respect to the original mesh non-orthogonality is merely 0.05%. These values ensured that all the simulations in the present campaign could be completed in fully automated fashion without crashes were reported or significant issues were observed. The results reported in the bottom right plot indicate that the effect of the mesh morphing algorithm proposed is that of increasing the grid average non-orthogonality values. This is somewhat expected, as the original volumetric grid in this work was generated making use of the snappyHexMesh tool of the OpenFOAM library. In such framework, most of the cells in the internal regions of the domain are substantially the result of an octree refinement of an original block mesh aligned with the coordinate axes. It is clear that the RBF procedure

---

2   Imposed for *y* symmetry conservation.

described in Section 2 does quite clearly alter in a non negligible way the orthogonal angles of a portion of the hexahedral cells produced by snappyHexMesh. Yet, the average increase in the average mesh non-orthogonality index is 2%, while the maximum increase observed is 7.2%, which are values that should not significantly affect the results of the simulations.

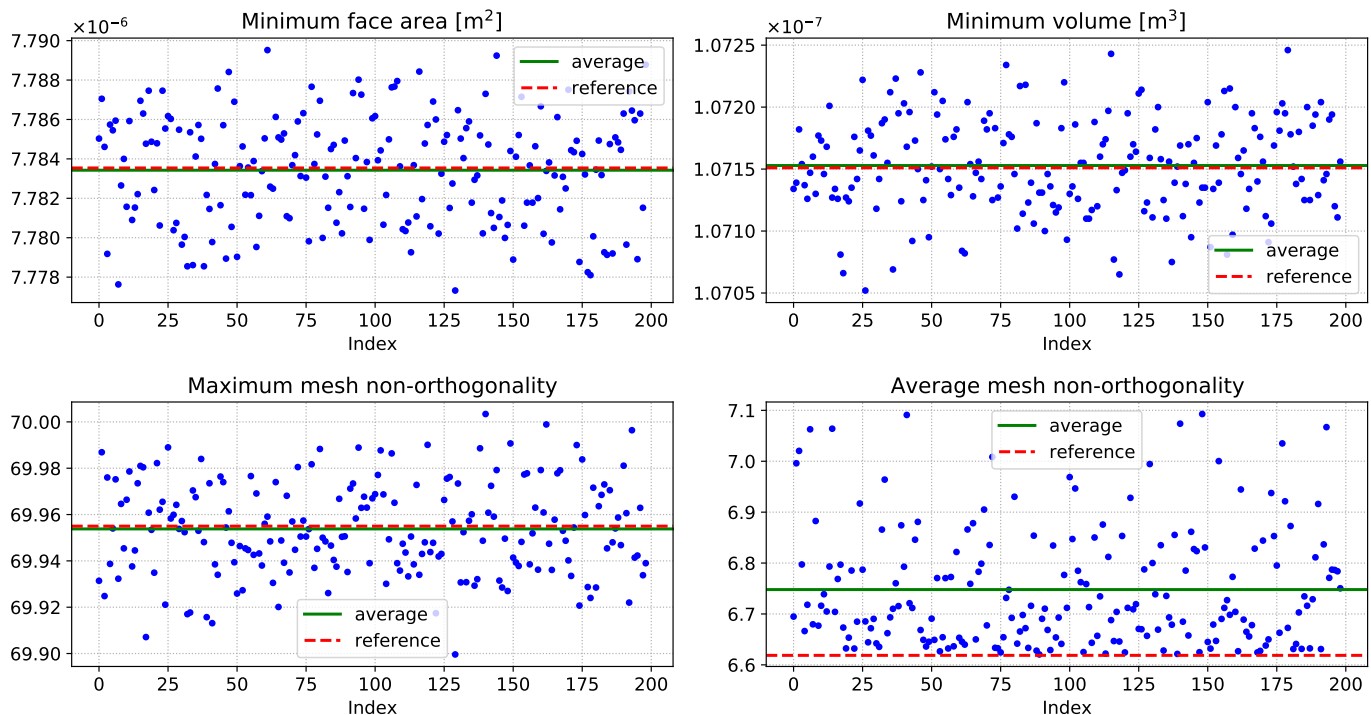

**Figure 8.** Values of the main mesh quality indicators as reported by checkMesh utility of OpenFOAM library, as a function of the index corresponding to each of the 200 volumetric meshes produced for the simulation campaign.

### 5.2. Reduced Order Model Construction

We set the full order model in scale 1:59.407, keeping it unaltered from the original work mainly for validation purpose. The computational domain, that is a parallelepiped of dimension $[-26, 16] \times [-19, 0] \times [-16, 4]$ along $x, y$ and $z$ directions is discretized in $8.5 \times 10^5$ cells, with anisotropic vertical refinements located particular in the free-surface region, in order to avoid a too diffusive treatment of the VOF variable. Boundaries of such domain are imposed as follows:

- at the *inlet* we set constant velocity, fixed flux condition for the pressure and a fixed profile for the VOF variable;
- at the *outlet* we set constant average velocity, zero-gradient condition for the pressure and variable height flow rate condition for VOF variable;
- at the bottom and lateral planes, we impose symmetric conditions for all the quantities;
- at the top plane, we set a pressure inlet outlet velocity condition for the velocity and nil pressure; VOF variable is fixed to 1 (air);
- at the hull surface, we impose no-slip condition for velocity, fixed flux condition for the pressure and zero-gradient condition for VOF variable.

The adopted solver is *interFoam*, which is able to solve the Navier Stokes equations for two incompressible, isothermal immiscible fluids. Time discretization uses a first order implicit scheme with local-step, since we are interested to the steady solution. For the spatial discretization, we apply a Gaussian integration using second order upwind scheme for divergence operators and linear interpolation for gradient and laplacian operator. By imposing a inlet velocity of 1.668 m/s, the Froude number is around 0.22. The time required

to converge to the steady solution within such setting on a parallel machine (32 processors) is approximately 2 h.

For the construction of the reduced order model, we randomly sample the parametric space with uniform distribution. We performed 203 simulations with the full order model, collecting the corresponding pressure and shear stress distributions (the latter implicitly containing the distribution of the VOF variable) over the hull surface. Thus, only the surface fields are considered at the reduced level. We then flatten the shear stress vector field in order to construct two snapshots matrices, one for the pressure and one for the stress. Both are then decomposed using POD technique. The number of modes considered is fixed to 20. Approximating the manifold with the GPR method, we obtain two different POD-GPR model that approximate the pressure field and the shear stress field. Such quantities are used for the computation of the objective function during the optimization procedure.

Even if the difference of hardware used for full order model simulations and for reduced order approximation limits the possible speedup obtained—a HPC facilities versus an ordinary personal computer—, we achieve satisfactory computational gain. In fact, whereas the FOM lasts approximately two hours, the ROM approximation only consisting in two distribution queries and two matrix multiplications, takes less than 1 s in a single-processor environment. Such results are very effective in the framework of an iterative process, as the optimization pipeline here proposed. The overall time is in fact mainly constituted by the initial FOM simulations needed for the offline database, while the ROM approximation can be considered negligible from the computational point of view. Moreover, it can be performed on significantly less powerful machines.

Adopting data-driven methodologies rather than projection-based ones has different advantages which we have already discussed, but shows also some drawback in the error bounding. For an a posteriori quantification of the ROM accuracy we need then to validate the approximated optimal result by carrying out a FOM simulation. We remark that we consider the output of such simulation as truth solution. This requires an additional computational cost, but allow also for an effective refinement of the ROM. Once a geometrical configuration is validated in such fashion, depending on the error observed we can add this last snapshot to the database and re-build the ROMs.

*5.3. Optimization Procedure*

We first define the objective function we applied to the optimization procedure. The quantity to minimize is the total resistance coefficient $C_t$, which is defined as

$$\min_{\mu} C_t \equiv \min_{\mu} \int_{\Omega(\mu)} \frac{\tau_x \rho - p n_x}{\frac{1}{2}\rho V^2 S}, \tag{15}$$

where $\tau_x$ is the $x$-component of the shear stress, $\rho$ is the fluid density, $p$ indicates the pressure, $n_x$ the $x$-component of the surface normal, $V$ and $S = \Delta^{2/3}$ the reference fluid velocity and the reference surface, respectively. As reported, the CFD simulations have been carried out in fixed sink and trim conditions. Thus, the specific reference surface used to obtain $C_t$ has been selected to penalize hulls obtaining resistance gains through immersed volume reduction. All the geometrical quantities, as well as the normals and the reference surface depend by the imposed deformation. Thus, to evaluate the $C_t$ for any design, we deform the hull surface using the FFD map, then project the ROM approximated fields—pressure and shear stress—on it to numerically compute the integral defined in Equation (15).

Regarding the ASGA hyperparameters, we set the probability of crossover and mutation as $P_C = P_M = 0.5$. For each solutions database we perform an optimization run with ASGA composed by 150 generations, with an initial random population of 100 individuals and an offspring of 20 individuals. The number of points returned by the AS back mapping is $B = 2$, while the dimension of the AS is set to 1 for every population. The covariance matrix for the active subspace computation is approximated using local linear models [45].

For each optimum found by ASGA we run a new high-fidelity simulation for validating the approximated $C_t$, adding the high-fidelity snapshots to the database in order to refine the POD-GPR model. In Figure 9 we show the comparison of all the runs. The third and last optimization reached a reduction of ∼1.4% of the $C_t$ coefficient compared to the original shape.

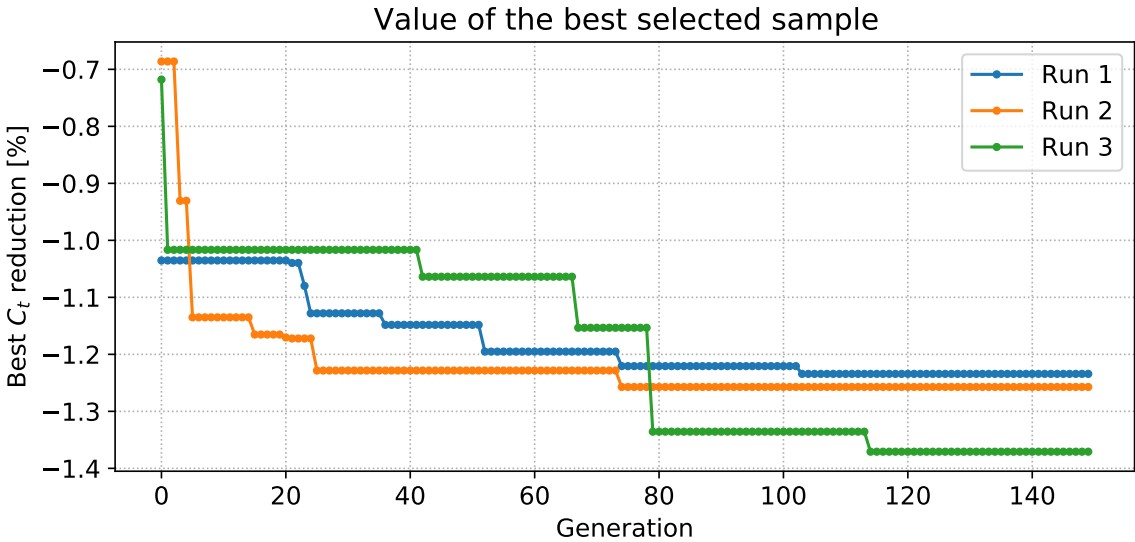

**Figure 9.** ASGA runs. The reduction of the $C_t$ is to be intended with respect to the undeformed reference hull.

Figure 10 presents the frontal sections of the optimal shape compared to the undeformed one, showing a volumetric increment in the frontal part which balances the reduction near the central zone. The a posteriori validation confirmed the positive trend: the $C_t$ coefficient of the optimal shape is 1.2% less, with a relative error of the ROM model of 0.18%. As is appreciable in Figure 10, the optimal hull has a wider section in the region immediately downstream with respect to the bulbous bow, while it appears slightly narrower in the middle ship sections. The immersed volume of the optimized hull is only 0.08% different from that of the original hull, which suggests that the $C_t$ reduction obtained is the result of a total resistance reduction. A possible interpretation of such a resistance decrease is that having a more streamlined hull along the longitudinal direction, is likely able to reduce the extent and dimension of the separation bubble located on the side of the bulbous bow, and corresponding to the dark blue strip visible in the wall shear stress contours presented in Figures 11 and 12. As a consequence, the optimal hull presents slightly lower pressures with respect to the original hull, in the region located downstream of the bulbous bow. Such a minimal reduction is hardly noticeable in the pressure contour plots presented in Figures 13 and 14. More appreciable differences are visible instead in the free surface elevation plot presented in Figure 15. Reducing the extent of the aforementioned detachment bubble, the shape modification leading to the optimal hull has the effect of moving forward the trough which follows the bow. This indicates that the pressures in the bow region are reduced, which results in a net decrease of the resistance pressure component. In fact, this leads to a 4.92% reduction in the pressure component of the resistance, against a more modest 0.55% reduction of viscous resistance. Yet, considering that the latter component accounts for approximately 83% of the total resistance, this translates into the 1.2% reduction reported. Finally, to exclude the possibility that the differences observed in the total resistance coefficient values are a result of possible discretization error due to the mesh morphing procedure, we report that the average and maximum values of wall $y^+$ of the optimized hull do not significantly differ from those obtained with the original one. The average and maximum wall $y^+$ values for the original hull simulation are 6.18426 and 99.5631, respectively, while the corresponding average and maximum values for the

optimized hull are 6.19071 and 99.6255, respectively. We point out that the $y^+$ maxima here reported for the DTC tutorial appear outside of the range prescribed for the turbulence model here used. Yet, the accuracy of the DTC tutorial results suggests that maxima $y^+$ is likely located outside the water. In fact, considering the small density of air with respect to water, the impact of the resulting inaccurate estimation of surface derivatives is minimal.

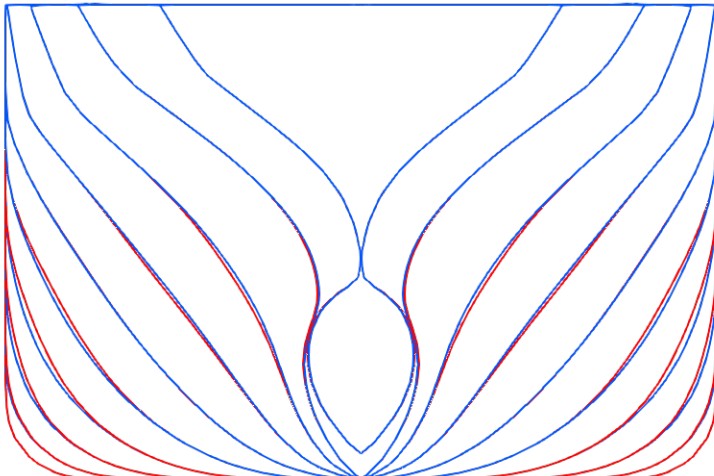

**Figure 10.** The sections (from 10 to 20) of the original ship in blue and of the optimized one in red.

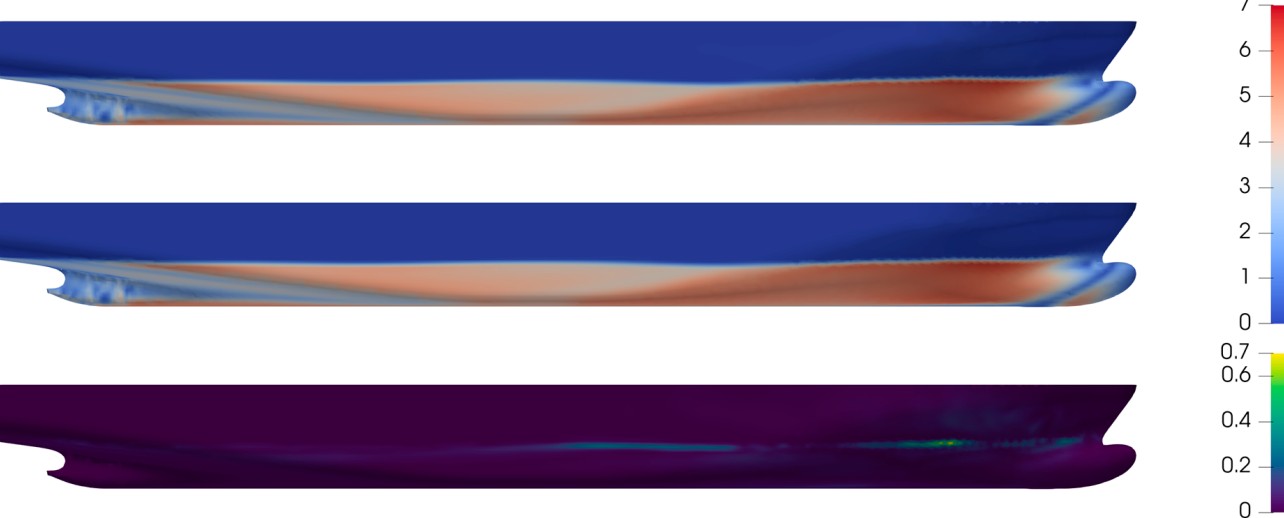

**Figure 11.** Distribution of the shear stresses measured in Pascal over the undeformed hull: the ull order model (FOM) validation (**top**) is compared to the reduced order model (ROM) approximation (**middle**) and the absolute error is shown (**bottom**).

We remark that the POD-GPR model approximates the distribution of the output of interest, not the objective function—which is computed using the predicted fields. For this reason, we can also compare the pressure and shear stresses over the optimal hull with respect to the undeformed one. Figures 11 and 13 present the graphical investigations about the ROM approximation error distribution over the undeformed hull, both for pressure and stresses distributions. For a more realistic comparison, we specify that the FOM snapshots referring to the undeformed geometry has been removed from the database, emulating the approximation any untested parameter. We proceed in the same way also for the optimal shape (Figures 12 and 14), not only to measure the accuracy of the POD-GPR model, but also for investigating the reasons of the $C_t$ reduction from a physical perspective. The absolute error is quite small, but it is possible to note that for both the fields it is mainly concentrated along the free-surface.

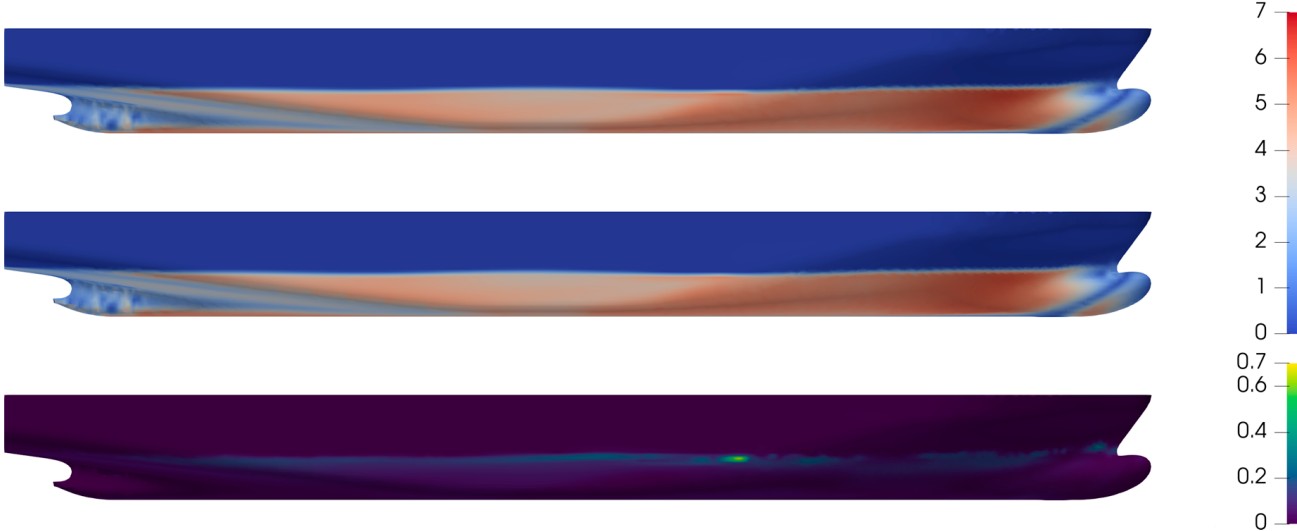

**Figure 12.** Distribution of the shear stresses measured in Pascal over the optimal hull: the FOM validation (**top**) is compared to the ROM approximation (**middle**) and the absolute error is shown (**bottom**).

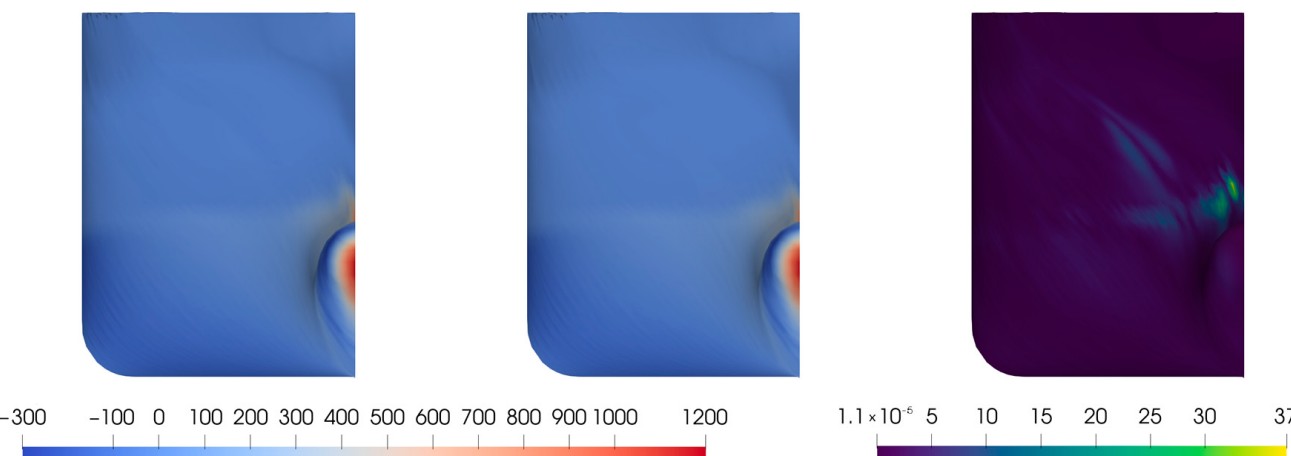

**Figure 13.** Distribution of pressure measured in Pascal over the undeformed hull: the FOM validation (**left**) is compared to the ROM approximation (**center**) and the absolute error is shown (**right**).

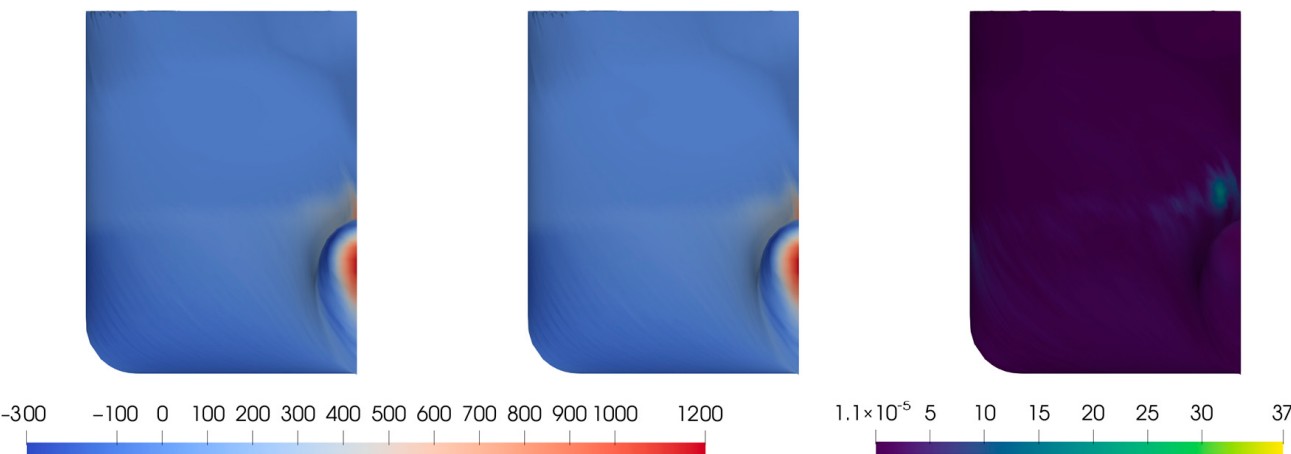

**Figure 14.** Distribution of the pressure measured in Pascal over the optimal hull: the FOM validation (**left**) is compared to the ROM approximation (**center**) and the absolute error is shown (**right**).

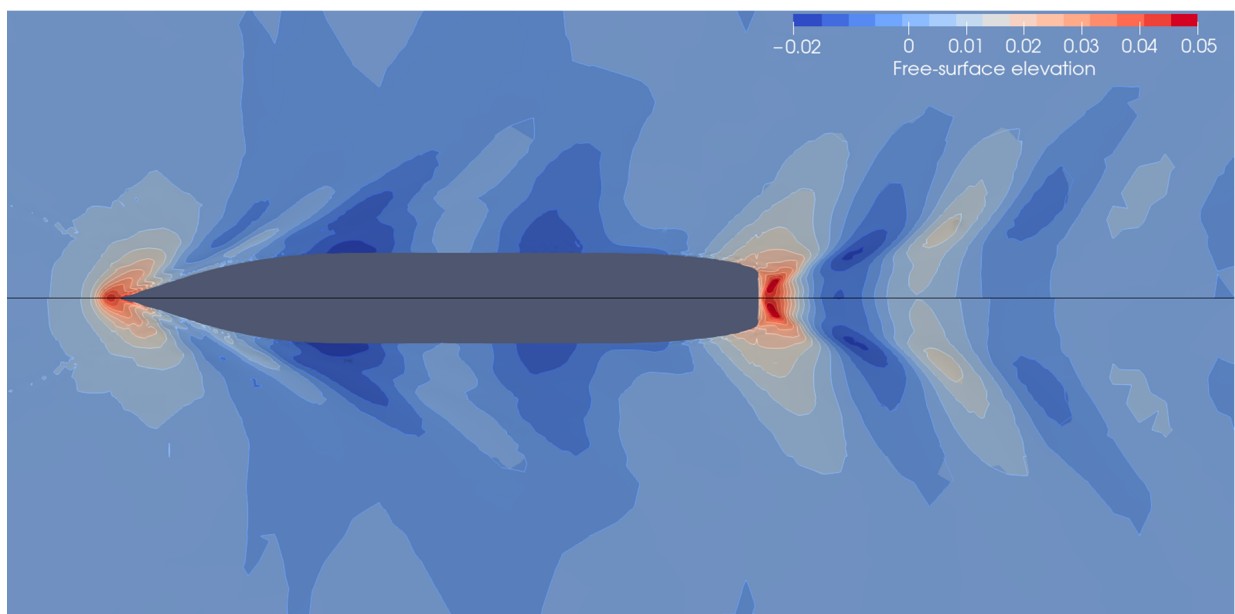

**Figure 15.** Contours of free surface elevation field around the original hull (**top** half) and optimal (**bottom** half).

Comparing the original hull with the optimal one we emphasize that the optimal shape seems to be able to slightly reduce the height of the wave created by its body, inducing a reduction of the wet surface. The friction resistance computed as the integral of the $x$ component of shear stresses over the two hulls shows in fact this marginal gain: the 12.76 N of the original ship becomes 12.69 N in the optimal configuration. However, the main contribution of the resistance reduction comes from the pressure resistance. While in the original shape we measure 2.64 N, in the optimized such quantity decreases to 2.51 N.

## 6. Conclusions

In this work we presented a complete numerical pipeline for the hull shape design optimization of the DTC benchmark hull. We proposed a self-learning geometrical deformation technique, where different morphing methods are coupled together to propagate surface deformations to volumetric meshes. Though in this work we used a FFD approach for the CAD modifications, we emphasize that our methodology can exploit any surface deformation. The optimization procedure is based on a coupling between active subspaces and genetic algorithm, called ASGA. For the evaluation of the total resistance coefficient for new untested parameters we exploits the non-intrusive data driven reduced order method called POD-GPR. This results in a great computational saving for the computation of the pressure and viscous forces fields, while preserving a good accuracy. We performed 3 optimization runs, with high-fidelity validation of the approximated optimum and enrichment of the solutions database to increase the accuracy of the ROM in its neighborhood. We obtained a reduction of the total resistance coefficient equal to 1.2% with respect to the original reference hull.

In the future, further investigations will be carried out to study a dynamic selection of the active subspace dimension, and a varying number of points returned by the back mapping procedure. Further improvements in the shape parameterization algorithms could be obtained improving the efficiency of the RBF weights computation. This could be obtained with a smarter selection of the RBF control points or, in a more invasive fashion, by resorting to fast algorithms—such as Fast Multipole Method—for the computation of the control points mutual distances.

**Author Contributions:** Methodology, N.D., M.T., A.M., G.R.; software, N.D., M.T., A.M.; investigation, N.D., M.T., A.M.; writing—original draft, N.D., M.T., A.M.; writing—review and editing, N.D., M.T., A.M., G.R.; visualization, N.D., M.T., A.M.; supervision, G.R. All authors have read and agreed to the published version of the manuscript.

**Funding:** This work was partially supported by an industrial Ph.D. grant sponsored by Fincantieri S.p.A., and partially funded by the project UBE2-"Underwater blue efficiency 2" funded by Regione FVG, POR-FESR 2014-2020, Piano Operativo Regionale Fondo Europeo per lo Sviluppo Regionale. It was also partially supported by European Union Funding for Research and Innovation— Horizon 2020 Program—in the framework of European Research Council Executive Agency: H2020 ERC CoG 2015 AROMA-CFD project 681447 "Advanced Reduced Order Methods with Applications in Computational Fluid Dynamics" P.I. Gianluigi Rozza.

**Institutional Review Board Statement:** Not applicable.

**Informed Consent Statement:** Not applicable.

**Data Availability Statement:** Not applicable.

**Conflicts of Interest:** The authors declare no conflict of interest.

**Abbreviations**

The following abbreviations are used in this manuscript:

| | |
|---|---|
| AS | Active subspaces |
| ASGA | Active subspaces genetic algorithm |
| CAD | Computer-aided design |
| CFD | Computational fluid dynamics |
| FFD | Free form deformation |
| FOM | Full order model |
| GA | Genetic algorithm |
| GPR | Gaussian process regression |
| HPC | High performance computing |
| PDE | Partial differential equation |
| POD | Proper orthogonal decomposition |
| POD-GPR | Proper orthogonal decomposition with Gaussian process regression |
| RBF | Radial basis functions |
| RANS | Reynolds averaged Navier–Stokes |
| ROM | Reduced order method |
| STL | Stereolithography tesselation language |
| VOF | Volume of fluid |

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
