# Peer review of "Hull Shape Design Optimization with Parameter Space and Model Reductions, and Self-Learning Mesh Morphing"

_jmse, doi:10.3390/jmse9020185_

Round 1

Reviewer 1 Report

Dear Authors,

In this paper, you studied the use of a non-intrusive POD based approach, using a mutlivariate Gaussian process method, all along with an active subspace genetic algorithm, in the purpose of searching efficiently to the optimal design parameters of a mesh morphing for fluid simulations that minimize a quantity of interest.

As stated in your paper, the originality of this work with respect to some precedent works, is the coupling of a surface deformation technique with the RBF approach in order to compute the volume nodes displacement.

The manuscript is very well written and is easy to read. This is very appreciated by the reviewer.

However, I have several remarks, questions and suggestions. It would be very appreciated if you could modify the manuscript with the answers to these questions.

1) Line 29: "to find the best shape.."

2) Line 47: "is related to the parametrization of the geometry."

3) Section 3.2, line 3: a reference number is missing.

4) Section 3.2:  does the GPR preseve the property of the modal coefficients, which states that they are not correlated ? (property of the Mercer theorem)

5) Line 329: could you please give the computational cost of the POD?

6) Line 338: could you please give the details of all the operations within the ROM approximation? 

7) Line 364: what is the reduced parametric space dimension obtained in association with the rank of the covariance matrix? what is the speed up with respect to the classical genetic algorithm?

Best regards.

Author Response

we attach a pdf file

Reviewer 2 Report

Thanks to the Authors for this interesting paper. The investigated topic about the hull shape design optimization using advanced techniques represents a relevant and noteworthy topic, especially for the industrial and design application.

However, as a general comment, the paper is not so easy to follow and a reshaping of the main paragraphs could be useful to make the paper easy to read.

Furthermore, the optimization pipeline and the procedures exposed in this paper are extremely similar to the paper “An efficient computational framework for naval shape design and optimization problems by means of data-driven reduced order modeling techniques” Demo, N.; Ortali, G.; Gustin, G.; Rozza, G.; Lavini, G.,  Bollettino dell’Unione Matematica Italiana 2020. Apparently, the main difference seems to be connected only to the tested hull (a cruise ship instead of a container ship).

The abstract seems not comprehensive on all topics addressed in the paper. I suggest modifying and extending it.

Other comments are the following:

  • Figure 3.

The caption could be reduced. This part s redundant “d) The displacements of the surface points are used to feed the RBF algorithm which computes the new position of the volumetric mesh nodes. We point out that to avoid distortion of the volumetric mesh symmetry plane, the surface mesh must include both sides of the hull.”

  • 3.2. The reduced order model: POD-GPR

Please check in this paragraph, there is a reference missed, please check it.

  • Please check the reference section, there are some references not fully complete. See for instance the following:

“Romor, F.; Tezzele, M.; Rozza, G. ATHENA: Advanced Techniques for High dimensional parameter spaces to Enhance Numerical Analysis. Submitted 2020.”

  • 5. Numerical results

The numerical paragraph needs to be extended. Firstly, I can suggest adding the details (dimensions and main parameters) of the DTC hull. Furthermore, could be relevant add the domain dimensions and the boundaries condition applied.

  • 5.3. Optimization procedure

I would recommend adding more details about the optimized configuration. Does the optimized hull shape give an advantage in terms of friction resistance or wave resistance? Can the Authors show the pressure plots on the optimized hull shape?

Author Response

we attach a pdf file

Reviewer 3 Report

The manuscript describes whole numerical procedure for the shape optimization adopting and combinating FFT, POD, ASGA, RBF and so forth. The methods and process are applied for the DTC hull and the resistance estimated by CFD simulations for the original and optimized hulls are compared. The topic of the paper may actually be interesting for the research community working in the same field.

As an overall opinion, the work is good but more details on the CFD simulation and its results should be addressed.

*Major comments

  1. Detailed information on the CFD simulation (condition) should be addressed such as boundary conditions, used solver (ex. InterFoam), snapshots of computational mesh near the hull, grid numbers, Froude Number and so.
  2. The reason for the reduction of *C_t due to shape change is needed such as comparisons of hull pressure and wave-height contours. Also, average and maximum y+ on the hull surface is needed to be presented.

*In the fields of naval architecture and ocean engineering, total resistance (R_t), which is summation of viscous and wave-making resistances, and non-dimensionalized of total resistance C_t is used instead of C_d.

  1. Hull form design including CFD simulations are routine but very time-consuming works. Please, compare and/or address the time-reduction of hull-form design and CFD simulations by adopting the present method and procedures. Computational environments, as well.

*Minor comments and suggestions

  1. title: “~space ‘and’ model reductions ‘and’ self-learning~”
  2. It is suggested to include the figure to illustrate the whole procedure of the research for the readers’ convenience.
  3. Some figure captions are too long and the same with the text (Ex. Figs. 2, 3 and 7). Would you like to make them more concise?
  4. Eq.(4): for the N.S. equations, external forces, e.g., pressure and viscous forces, are generally placed at the RHS of the Equation. Gravity is also needed to be placed since the present simulation considers free-surface.
  5. Regarding Major comment 2, it is recommend that the authors decompose and compare the resistance components following ITTC 1978 procedure in the authors’ future work.
  6. If proper, refer the following study.

JEONG, Kwang-Leol; JEONG, Se-Min. A Mesh Deformation Method for CFD-Based Hull form Optimization. Journal of Marine Science and Engineering, 2020, 8.6: 473. https://www.mdpi.com/2077-1312/8/6/473

Author Response

we attach a pdf file

Round 2

Reviewer 2 Report

Thanks to the Authors for the applied changes, the paper has been strongly improved. I have only a few further comments.

As a general comment, I would suggest checking the abbreviations, seems that there are some abbreviations missed.

3.1. The full order model: incompressible RANS

“In this work, we adopt a consolidated FOM, replicating the one discussed in [21]. Such model is nowadays widespread in many industrial naval contexts thanks to its robustness. Here we provide a minimal description. We refer to the original work for all the numerical and technical details.”

What means this sentence? The setup adopted for the Full Order Model was derived from El Moctar’s paper? Do the Authors mean that the simulation setup (exposed in that paper) is nowadays widespread? But as far as I recall that paper talks about mainly the experimental results and only the last part exposes an example of a comparison between experimental and simulation results. Furthermore, I would say that the DTC is not the widest benchmark hull for container vessels. Indeed, the most widespread is the KCS (KRISO Containership).

  1. Numerical results

Can the Authors provide the verification and validation analysis of the original hull shape? The verification analysis could be relevant for the estimation of the mesh uncertainty (mainly) and gives an idea of the reliability of the initial simulation setup (of course is clear that this is not strictly useful for the optimization procedures exposed).

Author Response

please look at the pdf attached

Reviewer 3 Report

Thank you very much for the authors' effort. 
Suggestions and minor comments are as follows. 

1.It is difficult to find the difference of hull surface pressure between original and optimal hulls (Fig.s 13-14). It is better to compare free-surface (elevation) contours.

2.Generally, y+ is checked to confirm grid size normal to the surface for the use of selected turbulence model for the CFD simulations. Please, recheck whether the text (line 409~414) is needed and/or proper.
*Although debates still remain, for the k-omega (SST) turbulence model, recommend y+ is under 20, ideally 1. 

Author Response

please look at the pdf

Round 3

Reviewer 2 Report

Dear Authors,

thanks again for the improvements applied to the paper. The paper has been enhanced. 

However, about the Author's answer to the comment related to the “Numerical results”, I would suggest the Authors check the ITTC 2017, 7.5-03-01-01 “Uncertainty Analysis in CFD Verification and Validation Methodology and Procedures”. This is the standard approach for the validation and verification in marine hydrodynamics applications. Indeed, no verification or validation has been provided in the quoted Master Thesis or also in the OpenFOAM library tutorials. So, can the Authors provide other references?

Except for this, I don’t have further comments.